

# Zooplankton as a potential vector for white band disease transmission in the endangered coral, *Acropora cervicornis*

Rebecca H. Certner[1,*], Amanda M. Dwyer[1,*], Mark R. Patterson[1,2] and Steven V. Vollmer[1]

[1] Department of Marine and Environmental Sciences, Northeastern University, Boston, MA, United States of America

[2] Department of Civil and Environmental Engineering, Northeastern University, Boston, MA, United States of America

[*] These authors contributed equally to this work.

## ABSTRACT

Coral diseases are a leading factor contributing to the global decline of coral reefs, and yet mechanisms of disease transmission remain poorly understood. This study tested whether zooplankton can act as a vector for white band disease (WBD) in *Acropora cervicornis*. Natural zooplankton communities were collected from a coral reef in Bocas del Toro, Panama. Half of the zooplankton were treated with antibiotics for 24 h after which the antibiotic-treated and non-antibiotic-treated zooplankton were incubated with either seawater or tissue homogenates from corals exhibiting WBD-like symptoms. A total of 15 of the 30 asymptomatic *A. cervicornis* colonies exposed to zooplankton incubated in disease homogenate in tank-based experiments showed signs of WBD, regardless of prior antibiotic incubation. These results indicate that in our experimental conditions zooplankton were a vector for coral disease after exposure to disease-causing pathogens. Given the importance of heterotrophy on zooplankton to coral nutrition, this potential mode of disease transmission warrants further investigation.

## INTRODUCTION

Coral diseases are on the rise across the world's reefs and are likely to spread and worsen with the growing impact of climate change (*Maynard et al., 2015*). However, despite their increasing threat to global coral health, coral disease etiologies often remain elusive. White band disease (WBD) is one such enigmatic condition. Since its initial detection in 1979, WBD has devastated populations of the Caribbean acroporids, *Acropora palmata* and *Acropora cervicornis* (*Aronson & Precht, 2001*; *Gladfelter, 1982*; *Bythell, Gladfelter & Bythell, 1993*). As a direct result of WBD outbreaks, both corals are now classified as critically endangered by the IUCN Red List of Threatened Species. Despite its profound effect on Caribbean reef ecosystems, key aspects of WBD causation and transmission remain unknown.

The WBD epizootic is highly infectious and has been shown to be transmissible through the water to injured corals, as well as through coral-coral contact (*Kline & Vollmer, 2011*).

Corresponding author
Rebecca H. Certner,
certner.r@husky.neu.edu

WBD is generally thought to be bacterial since it can be impeded by antibiotics and stimulated by exposure to bacterial autoinducers (*Kline & Vollmer, 2011*; *Sweet, Croquer & Bythell, 2014*; *Certner & Vollmer, 2015*). Although the exact consortium of pathogens has not been identified, several bacterial families are thought to be involved in WBD infection including Vibrionaceae, Alteromonadaceae, and Flavobacteriaceae (*Gil-Agudelo, Smith & Weil, 2006*; *Sweet, Croquer & Bythell, 2014*; *Gignoux-Wolfsohn & Vollmer, 2015*). Waterborne transmission of WBD has also been established in tank-based experiments (*Gignoux-Wolfsohn, Marks & Vollmer, 2012*; *Certner & Vollmer, 2015*). Coral disease is also spread by corallivores such as fireworms, snails, and a variety of reef fish (*Sussman et al., 2003*; *Gignoux-Wolfsohn, Marks & Vollmer, 2012*; *Rotjan & Lewis, 2008*).

Diverse marine bacterial communities play vital roles in ocean ecosystems including nutrient cycling, decomposition of organic matter, and acting as a basal food source level (*Das, Lyla & Khan, 2006*). Marine bacteria are commonly found attached to various surfaces including zooplankton (*Karner & Herndl, 1992*). For example, copepod exoskeletons and guts are rich in bacteria (*Tang et al., 2011*). The chitinous exoskeleton of zooplankton may provide stable microhabitats for bacterial colonization (*Karner & Herndl, 1992*). Zooplankton have been demonstrated to be reservoirs for marine pathogens—specifically *Vibrio cholerae*—implicated in human diseases (*Tamplin et al., 1990*). Gammaproteobacteria, the bacterial class containing the greatest number of known pathogens, dominates copepod-associated bacterial communities, especially compared to the surrounding seawater (*Shoemaker & Moisander, 2015*). Within the Gammaproteobacteria class, Vibrionaceae, Alteromonadaceae, and Pseudoalteromonadaceae were found to be particularly common families associated with zooplankton, all of which are also linked to WBD in corals (*Gil-Agudelo, Smith, & Weil, 2006*; *Gignoux-Wolfsohn & Vollmer, 2015*). In addition, zooplankton have been shown to be vectors impacting marine organisms including fish and shellfish as zooplankton are suspected reservoirs for the highly infectious birnaviruses (*Kitamura et al., 2003*).

In this study, we investigated whether zooplankton facilitate the spread of WBD when asymptomatic *A. cervicornis* are exposed to "WBD-infected zooplankton" and thus explore the potential connection between coral heterotrophy and waterborne disease transmission.

## MATERIALS AND METHODS

### Experimental design and field methods

Sixty asymptomatic *A. cervicornis* fragments were collected from Coral Cay (9°15′16″N, 82°7′40″W) in Bocas del Toro, Panama in February 2016. Each 8 cm fragment was collected using clippers from a distinct coral colony showing no tissue loss. Corals were at least two meters apart to remove any effect of colony genotype. Collection permits were provided by Autoridad Nacional del Ambiente (ANAM) SE/A-9-16, Republic of Panama. Coral fragments were acclimated in 12 aquaria for 48 h in a flow-through system. During this time, all corals remained visually healthy. Zooplankton were collected in three separate 10 min tows by a diver with a 50 $\mu$m plankton net directly over *A. cervicornis* patches at 7 m depth. From each tow, three subsamples of zooplankton were counted to estimate total

zooplankton collected; the communities were dominated by copepods and polychaetes. All zooplankton were then concentrated into 600 ml of 50 μm-filtered seawater (50 μm FSW) and inverted to ensure homogeneity before dividing evenly among 12 bottles (50 ml bottle$^{-1}$). A total of 50 μm FSW was then added to each bottle up to 200 ml. We estimate that there were ∼90 zooplankters bottle$^{-1}$. An antibiotic cocktail containing tetracycline, ampicillin, and chloramphenicol was added to six of the aliquots at 100 μg ml$^{-1}$ and all aliquots were incubated for 24 h in the dark. After 24 h, 75 μl from each aliquot was plated onto LB media (general culture media made with seawater) and TCBS media (*Vibrio*-selective media) in order to determine antibiotic efficacy. After 24 h zooplankton could be seen actively swimming in all bottles.

The following day, 18 WBD-infected *A. cervicornis* fragments were collected from the same reef and homogenized in 0.2 μm-filtered seawater (0.2 μm FSW). Six 30 ml WBD pools were created by combining 10 ml from three homogenates. WBD pools were spun at 500 rpm for five min to remove debris. Samples from each WBD pool were preserved for subsequent 16S rRNA gene sequencing in DNA buffer (*Fukami et al., 2004*). Each of the 12 aliquots of zooplankton was filtered at 0.2 μm. Zooplankton were rinsed from filter with 0.2 μm FSW into 12 new aliquots of 50 ml 0.2 μm FSW. Six aliquots (three plus-antibiotic and three minus-antibiotic) were dosed with a 30 ml WBD pool while the remaining six were supplemented with 30 ml 0.2 μm FSW, creating two levels of antibiotic exposure (plus antibiotic or minus antibiotic) that were fully crossed with two levels of disease exposure (plus WBD or minus WBD). This resulted in four treatments: (1) plus antibiotic, minus WBD, (2) plus antibiotic, plus WBD, (3) minus antibiotic, minus WBD, and (4) minus antibiotic, plus WBD. Zooplankton samples were incubated for 24 h, after which zooplankton could be seen actively swimming in all bottles.

Acclimated coral fragments were given a ∼7.5 mm$^2$ lesion using an airbrush with 0.2 μm FSW to mimic injury in the field known to increase susceptibility to WBD infection (*Gignoux-Wolfsohn, Marks & Vollmer, 2012*; *Certner & Vollmer, 2015*). Five corals were placed in 12 closed-system aquaria containing 4 l seawater each. A circulating powerhead in each aquarium was covered with 50 μm mesh to prevent damage to the zooplankton. Each of the 12 zooplankton aliquots (three replicates per treatment) were filtered with 50 μm mesh. The zooplankton were then thoroughly rinsed with 0.2 μm FSW and then transferred to aquaria using 0.2 μm FSW (Fig. 1). Corals were checked every 3 h for 96 h for WBD-like signs. Upon detection, the diseased fragment was removed from the aquarium and transferred to a holding aquarium to prevent contamination of other replicates. After 96 h, corals were removed and aquaria water was filtered through 50 μm mesh to collect remaining zooplankton. The mesh was then rinsed with 0.2 μm FSW directly into labeled dishes. Each dish was examined under a dissecting microscope to determine the number of remaining zooplankton.

## 16S library preparation and bioinformatics

Total DNA was extracted from disease pool samples using the BioSprint 96 DNA Blood Kit with the addition of PEB buffer (Qiagen, Hilden, Germany). Primers were created to target the V3–V4 hypervariable region of the 16S rRNA gene (Integrated DNA Technologies,

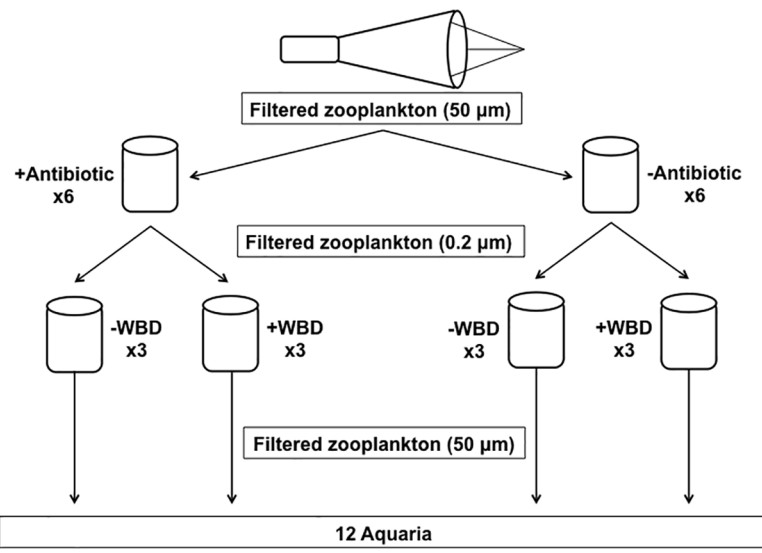

**Figure 1  Experimental design.** Antibiotic-treated and control zooplankton were incubated with WBD homogenates and seawater. Aquaria containing acclimated *A. cervicornis* were then dosed with one of these four treatments.

*Fadrosh et al., 2014*). PCR products were normalized and the resulting multiplexed paired-end libraries were sequenced on the Illumina 2,500 HiSeq platform at Tufts University (*Fadrosh et al., 2014*). Since sequences did not sufficiently overlap, single-read sequences (comprising the V4 region) were demultiplexed using a custom script. OTUs were clustered at 97% similarity using the open reference picking script from QIIME and UCLUST against the SILVA database (*Caporaso et al., 2010*; *Quast et al., 2013*). OTU taxonomy was assigned using BLAST against the SILVA database. OTU counts were normalized using the arithmetic mean modification as per *Gignoux-Wolfsohn & Vollmer (2015)* and *Anders & Huber (2010)*.

## Statistical analyses

We used a generalized linear model (GLM) with a binomial error distribution that accounts for overdispersion in order to determine whether the proportion of infected *A. cervicornis* fragments varied across the antibiotic (plus or minus) and disease (WBD or FSW) treatments using the MASS package (*Venables & Ripley, 2003*) for R. Statistical significance was determined via Likelihood Ratio Tests (LTR) using the R package car (*Fox & Weisberg, 2011*). We verified that the model residuals were normally distributed (Shapiro–Wilk test; $p = 0.251$) and homoscedastic (Levene's test; $p = 0.596$).

## RESULTS

Our post-antibiotic exposure plating results showed the antibiotic treatment was highly effective at killing bacteria as plates from the plus-antibiotic treatment had zero colonies present and plates from the minus-antibiotic treatment showed hundreds of colonies present. Zooplankton exposed to antibiotic resulted in zero CFUs on both LB and TCBS media for each aliquot. Conversely, non-antibiotic treated zooplankton resulted in plates containing hundreds of CFUs.
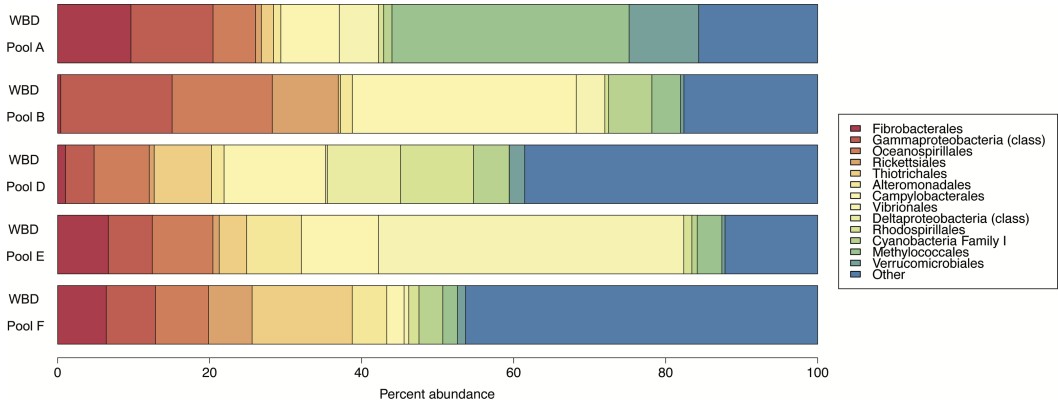

**Figure 2** **Relative abundance of 16S rRNA gene sequences for the WBD pools.** The percent abundance of taxonomic orders whose incidence reached five percent of the total OTU count in at least one of the five samples is shown. Rare taxonomic orders are grouped as 'Other'. WBD Pool C was not included in the figure due to a low number of sequences. WBD Pools A–C were added to minus antibiotic zooplankton aggregates and WBD Pools D–F were added to plus antibiotic zooplankton aggregates.

The six WBD pools were sequenced for 16S and 97% clustering yielded 1,919 OTUs. One of the six pools had an insufficient level of read depth and thus was excluded from the analyses. Disease pools were dominated by Proteobacteria, particularly Gammaproteobacteria (Oceanospirillales, Thiotrichales, Alteromonadales, Vibrionales, Methylococcales) and Alphaproteobacteria (Rickettsiales, Rhodospirillales) (Fig. 2). A large portion of the community for each of the WBD pools was dominated by low-abundance species labeled as "Other" (Fig. 2).

Corals exposed to zooplankton incubated in the WBD pools showed a significant increase in WBD infection (15 out of 30 fragments) compared to corals exposed to zooplankton incubated in FSW where no transmission occurred (Figs. 3 and 4). Our two-factor aquaria-based experiment showed a significant effect of disease exposure ($\chi^2(1) = 5.407$, $p = 0.02$), but not antibiotic exposure ($\chi^2(1) = 0.680$, $p = 0.41$) or an interaction on WBD infection in *A. cervicornis* ($\chi^2(1) = 0$, $p = 1$) (Fig. 4). Upon termination of the experiment, aquaria water was filtered at 50 µm to determine the state of remaining zooplankton. Fewer than 25% of the added zooplankton were recovered and the majority of the individuals from each aquarium were living.

## DISCUSSION

16S analyses were conducted in order to confirm that our WBD pools contained disease-causing bacteria. Our WBD pools contain many bacterial orders shown to be associated with coral disease in general and WBD specifically. Vibrionales has been linked to WBD on numerous occasions (*Certner & Vollmer, 2015*; *Gignoux-Wolfsohn & Vollmer, 2015*; *Gil-Agudelo, Smith & Weil, 2006*; *Ritchie, 2006*) as well as Rickettsiales (*Casas et al., 2004*), Oceanospirillales (*Gignoux-Wolfsohn & Vollmer, 2015*), Alteromonadales (*Gignoux-Wolfsohn & Vollmer, 2015*), and Campylobacterales (*Gignoux-Wolfsohn & Vollmer, 2015*) (Fig. 2). These results suggest that the zooplankton were incubated with likely WBD

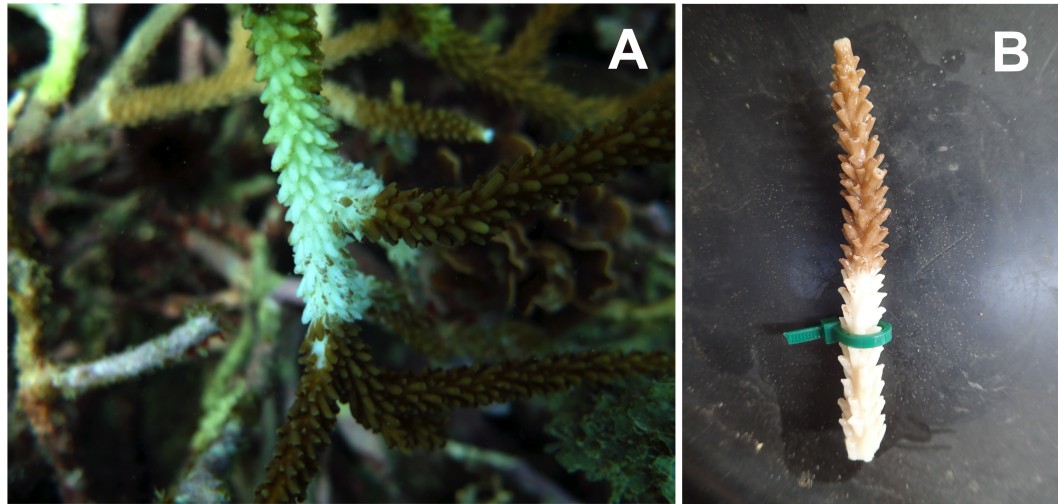

**Figure 3** **Examples of white band disease.** (A) WBD-infected *A. cervicornis* colony from Bocas del Toro, Panama. (B) Experimentally transmitted WBD on an *A. cervicornis* fragment from the experiment.

pathogens. Interestingly, Gammaproteobacteria—the principal taxonomic class in the disease pools— are strongly linked to bacterial colonization of zooplankton (*Heidelberg, Heidelberg & Colwell, 2002*; *Shoemaker & Moisander, 2015*). Multiple studies have found that zooplankton microbiomes are dominated by Gammaproteobacteria, especially *Vibrios*. These opportunistic species likely benefit from the nutrient-rich zooplankton microhabitat unavailable in seawater. As a result, it is probable that zooplankton are hospitable to WBD-associated microbes, generally thought to be within the Gammaproteobacteria class. In essence, the bacteria that zooplankton are most likely to harbor are the same bacteria most likely to cause disease in a variety of marine organisms including corals.

This study is the first to explore disease transmission via coral heterotrophy. Corals across all treatments were found to have consumed over 75% of the supplied zooplankton. This result indicates that asymptomatic corals feeding on zooplankton covered with disease-associated bacteria can become infected with WBD. Although incubating zooplankton with the WBD-associated microbiome is a contrived scenario, these results imply that, under certain conditions, zooplankton may act as a coral disease vector.

Zooplankton communities on coral reefs are extremely variable (*Heidelberg et al., 2010*). Therefore, it is difficult to assess the exposure of corals to zooplankton harboring disease-associated bacteria in the field. However, our results show that zooplankton act as reservoirs for WBD-associated bacteria thus facilitating the exposure of corals to concentrated infectious agents. We hypothesize that during WBD outbreaks, when infected coral tissue enters the surrounding environment at higher concentrations, zooplankton could come into contact with disease-causing bacteria at a local scale (*Haas et al., 2013*). Therefore, we suggest that zooplankton act as a vector for WBD, explaining how isolated disease outbreaks can spread quickly across reefs.

Further investigation is needed to better understand the mechanisms of zooplankton acting as a vector for WBD and its prevalence under natural circumstances. Although

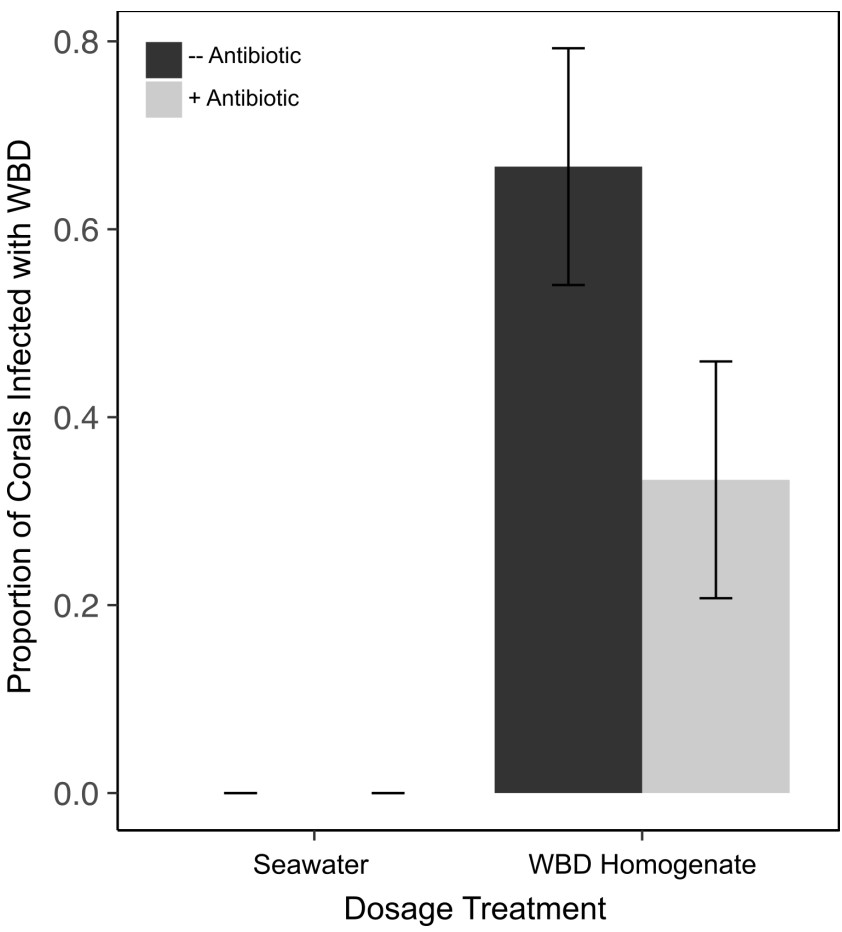

**Figure 4** **Incidence of WBD infection in asymptomatic *A. cervicornis* exposed to zooplankton incubated with WBD-associated bacteria.** *A. cervicornis* were dosed with one of the four zooplankton treatments: (1) minus antibiotic, minus WBD, (2) plus antibiotic, minus WBD (3) minus antibiotic, plus WBD, and (4) plus antibiotic, plus WBD. WBD was equated with the appearance of a distinct band of necrotic tissue. Incidence of infection was analyzed using ANOVA for significance of GLM terms. Mean ± SE shown.

lesioning corals is a standard procedure used to ensure WBD transmission in aquaria, we hypothesize that consumption of disease-associated bacteria via zooplankton heterotrophy may also infect uninjured corals. To further investigate the prevalence of zooplankton as disease vectors in the field, demersal plankton traps should be placed over WBD-infected *A. cervicornis* colonies. Comparing the bacterial communities of these zooplankton and the WBD-infected corals may reveal common OTUs, which would indicate that zooplankton spread WBD.

Zooplankton comprise the base of secondary production in most aquatic ecosystems and are a crucial component of innumerable food webs. However, negative impacts of zooplankton are understudied, especially in regards to disease transmission to other marine organisms. Here we show that zooplankton can act as a vector for coral disease, specifically WBD in *A. cervicornis*, under certain experimental circumstances where zooplankton are artificially exposed to high densities of disease-causing bacteria.

## ACKNOWLEDGEMENTS

We thank Sarah Gignoux-Wolfsohn, Erik Holum, and Tarik Gouhier for bioinformatics and statistical guidance, Francis Choi for valuable comments, and the gecko in the dryer. We also thank the Smithsonian Tropical Research Institute for field and lab support. Supported by National Science Foundation Awards #1458158 (SVV) and #1412462 (MRP), and Northeastern University. This is contribution number 355 from the Marine Science Center at Northeastern University.

### Funding

This work was supported by National Science Foundation Awards #1458158 and #1412462, and by Northeastern University. The funders had no role in study design, data collection and analysis, decision to publish, or preparation of the manuscript.

### Grant Disclosures

The following grant information was disclosed by the authors:
National Science Foundation: #1458158, #1412462.
Northeastern University.

### Competing Interests

The authors declare there are no competing interests.

### Author Contributions

- Rebecca H. Certner and Amanda M. Dwyer conceived and designed the experiments, performed the experiments, analyzed the data, contributed reagents/materials/analysis tools, wrote the paper, prepared figures and/or tables, reviewed drafts of the paper.
- Mark R. Patterson and Steven V. Vollmer conceived and designed the experiments, contributed reagents/materials/analysis tools, reviewed drafts of the paper.

### Field Study Permissions

The following information was supplied relating to field study approvals (i.e., approving body and any reference numbers):

Collection permits were provided by Autoridad Nacional del Ambiente (ANAM) SE/A-9-16, Republic of Panama.

### Data Availability

The fasta files for the WBD pools are accessible via FigShare (DOI: 10.6084/m9.figshare.4828669) and in the Supplemental Information.

### Supplemental Information

Supplemental information for this article can be found online at http://dx.doi.org/10.7717/peerj.3502#supplemental-information.

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
