# Peer review of "Zooplankton as a potential vector for white band disease transmission in the endangered coral, Acropora cervicornis"

_PeerJ, doi:10.7717/peerj.3502_

## Round 0.1 · original submission · Minor Revisions

· Academic Editor

Minor Revisions

Both reviewers agree that there needs to be more detail given in the Introduction and Methods, and both ask for additional evidence (photographic) of disease progression. The first reviewer asks some questions about replicates, but the second is more direct asking if there is any additional replicate analysis done. The study is important and interesting, but these issues should be addressed as best possible to better ensure readability and correctness.

Reviewer 1 ·

Basic reporting

Meets the standards of basic reporting

Experimental design

Question is well defined. Methodology is appropriate

Validity of the findings

Conclusions follow from the results and are linked to the original research question

Additional comments

Here are a few suggestions for the authors
1. Line 60. The vast majority of marine bacteria are certainly not attached to surfaces and zooplankton, but are free-living. I’d reword to something like “…vital roles in ocean ecosystems, and are commonly found attached…”

2. Lines 119-120. I know nothing about WBD or coral disease in general, but the idea that coral will show WBD-like signs in 96 hours seems surprising. Given that the key data in this study is WBD-like signs of disease, some additional detail on what these signs are would be helpful. Can the authors include some photographic evidence of disease progression?

3. Line 129. Why is there a reference to Fadrosh 2014 here? Is this where the primer sequences are reported? If so, what makes them custom? Some clarification here is warranted

4. Line 152. What is LB and TCBS media? And why use both? Some rationale should be presented here or in the methods.

5. Lines 170-182 and Figure 3. If I understand, these are 16S rRNA gene datasets from the homogenized WBD-infected coral. Are these replicates? If so, why are they so different. For example, about 40% of Pool A is comprised of Methylococcales, while Pool E is about 40% Vibrionales. Overall, the 16S rRNA data adds little to the study. Perhaps if the researchers had used 16S rRNA sequencing in a comparative fashion. For example, how does the bacterial community change on the coral as the WBD-like signs of disease progress? Since these data represent the inoculum used on the zooplankton, maybe it’s better to present them prior to the results on the incidence of disease. Or just get rid of them all together…

6. Figure 3. There’s no such thing as “16S community”. Please refer to these data as “16S rRNA gene sequences”.

·

Basic reporting

In general, this is an interesting and well-written manuscript. I think that it can be improved with further explanation and clarification. Some more specifics are required.
I would provide more details in the Abstract—for example, provide the sample number instead of just the percentage. Also the Abstract implies that the research was done in “Caribbean acroporids” when in fact only one species was examined.
The words “healthy” and “health” should be avoided. For example, we are told on line 83 that 60 healthy A.c. fragments were collected. It is not possible to tell if these fragments were healthy or not. I think they mean that visually there were no obvious signs of tissue loss—and that would be the way to describe this.
Mention the original citation for WBD by W. Gladfelter—not just Aronson and Precht 2001.
Line 60—What are some of the vital roles that marine bacterial communities play in ocean ecosystems??
Lines 62-64—this statement is quite confusing. How distinct are these microbes? Where would they be found? On copepods for example or in the water column? In other words, it is difficult to sort out the differences between the microbes on and in the zooplankton and those in the surrounding water and the different roles they might play.
Line 156—Figure 2 needs to be revised to make it clear that zooplankton were included in the seawater treatments.

Lines 177 to 178 This statement is not clear to me.
It would have been good to explore the link between these lab-based results and occurrences of WBD in the field. For example, has WBD been documented in areas where the pathogenic bacteria are more likely to be found in high concentrations?
How likely is zooplankton to be a factor in WBD transmission under natural conditions? This question needs to be addressed head-on and not just in lines 165 to 168 and in the final sentence of the manuscript.

Experimental design

The methods are not described in sufficient detail and with enough information to replicate the study. How exactly were the fragments collected? Did each fragment have a base that was already dead or were living fragments broken off of longer live branches? How big were the fragments?
How many colonies were the sources of these fragments and how far apart were they from each other?
It would be better to know the genotypes of these source colonies. Were they likely all one genotype? That could influence the results.
What was the reason for damaging the fragments—in other words, why was it necessary to “mimic injury in the field”? No rationale is provided, and this process (or its implications) is never discussed again in the paper. I am aware that some researchers have found that disease is transmitted more readily when corals are injured, but that is not the subject of this experiment.

Validity of the findings

It would be advisable to repeat the entire experiment when (if) feasible.
Line 120 refers to WBD signs, but these are only described (briefly) in a figure legend. I think that photographs should accompany this paper if there is a decision to publish it. Can these researchers be certain that the fragments were exhibiting white band?

Additional comments

I think it would strengthen the paper to include a section on Future Research—exploring additional experiments that could be done to help put this study in context.

---

## Round 0.2 · accepted · Accept

· Academic Editor

Accept

The authors have met the reviewers requests as best possible.